# Perspectives on the implementation of post-validation surveillance for lymphatic filariasis in the Pacific Islands: A nominal group technique-based study protocol

Adam T. Craig[1]*, Harriet Lawford[1], Satupaitea Viali[2,3], George Tuitama[2], Colleen L. Lau[1]

1 UQ Centre for Clinical Research, The University of Queensland, Brisbane, Queensland, Australia,
2 Oceania University of Medicine Samoa, Apia, Samoa, 3 National University of Samoa, Apia, Samoa

* adam.craig@uq.edu.au

## Abstract

### Introduction

Lymphatic filariasis (LF) is a neglected tropical disease caused by parasitic worms, transmitted via mosquito bites. Significant global efforts have led to the interruption of LF transmission, with eight of the 16 previously endemic Pacific Island countries and territories (PICTs) validated by WHO as having eliminated the disease as a public health problem. Post-validation surveillance (PVS) is recommended to verify the absence of a resurgence in transmission; however, there are no guidelines on how to implement such surveillance effectively.

### Aim

This protocol outlines a research study that aims to explore, synthesise, and prioritise the perspectives of LF program staff from LF-eliminated PICTs. The study will focus on identifying the challenges faced in implementing PVS in the PICT settings and determining corresponding context-relevant operational research priorities.

### Method

The study will employ a nominal group technique (NGT) involving representatives from LF-eliminated PICTs. The NGT will adhere to a structured process for generating, ranking, and prioritising ideas, followed by group discussions, debates, and the collation of shared views.

### Discussion

The research will provide insights into the most significant challenges faced in implementing PVS in the PICTs. It will also identify priority areas where operational research is required to inform policy and practice. To our knowledge, this study is the first to apply a rigorous consensus group method to distil challenges and research priorities for LF PVS in PICTs. We expect that this research will inform the development of national and regional LF guidelines.

**Data Availability Statement:** No datasets were generated or analysed during the current study. All

relevant data from this study will be made available upon study completion.

**Funding:** The author(s) received no specific funding for this work.

**Competing interests:** The authors have declared that no competing interests exist.

## Introduction

Consensus group methods are widely used to synthesise expert opinions when evidence is insufficient or conflicting. These methods have been applied across multiple disciplines, including healthcare, education, engineering, and management [1, 2], to inform a variety of health-related activities. These include defining diagnostic criteria, classifying diseases, selecting quality indicators, informing management guidelines, and educating healthcare professionals [1–5].

Nominal group technique (NGT) was originally developed for effective group decision-making within social psychological research [3]. The NGT follows a structured format to generate, rank, or prioritise ideas. The process typically involves presenting a nominal question; participants generating and reporting of ideas; group clarification, debate, and discussion; ranking and voting; and results and conclusion generation [6]. NGT is recognised for adhering to the foundational principles of consensus decision-making methods [2].

One distinctive feature of NGT is its emphasis on structured discussion and debate, which allows diverse ideas on shared interests to be expressed, explored, and collated. Where differences in opinion arise, the reasons for these can be discussed. This collaborative approach has been suggested to enhance stakeholders' sense of ownership over the resulting research and increase the likelihood of influencing practice or policy change [3]. While NGT's structured nature and engagement are key advantages, its limitations include the relatively small number of participants and the potential for dominant voices to influence group decisions.

Lymphatic filariasis (LF) is a neglected tropical disease (NTD) caused by three species of filarial worm: Wuchereria bancrofti, Brugia malayi, and B. timori and transmitted between humans by a range of mosquito species including Aedes, Culex, and Anopheles [7, 8]. W. bancrofti is the only filarial worm species found in the Pacific Islands [9]. Adult worms reside in the lymphatic system and can cause chronic damage over time if left untreated.

Lymphatic filariasis has affected about 882 million people, primarily in tropical and subtropical regions of Africa, Southeast Asia, the Pacific Islands, and the Americas [10]. As of 2018, 51 million people were infected—a 74% decline since the start of WHO's Global Programme to Eliminate Lymphatic Filariasis in 2000 [11]. Since 2000, over 9.7 billion treatments for LF have been administered through mass drug administration (MDA) campaigns, reaching more than 943 million people. In 2023, the number of people that require MDA was 657 million [12].

The Pacific Islands include 22 small island countries and territories located over one-third of the Earth's surface and are home to around 15 million people [13]. In 2000, the Global Programme to Eliminate LF (GPELF) was launched by the World Health Organization (WHO). The GPELF's main objective was to achieve the elimination of LF as a public health problem across 73 endemic nations, including 16 in the Pacific Islands region, by 2020 [14, 15]. Through multiple rounds of MDA over many years, eight previously endemic Pacific Island Countries and Territories (PICTs) have reduced the prevalence of LF to below the microfilaria-positive prevalence threshold of 1% where transmission is thought to be no longer sustainable [16]. The WHO has validated these countries as having achieved the elimination of LF as a public health problem. Eight other PICTs have yet to eliminate LF, and six are considered non-endemic for LF (Table 1) with no recorded evidence of LF transmission [12].

Countries that have received WHO validation of elimination of LF as a public health problem are encouraged to conduct periodic post-validation surveillance (PVS) to ensure that LF transmission has not been re-established; however, to date, there are no official WHO guidelines on how best or how frequently to conduct PVS [8].

**Table 1. Pacific Island country and territory lymphatic filariasis endemicity and elimination status (2024).**

| Pacific island countries and territories (PICTs) | Year validated by WHO as having eliminated LF | Mid-year population estimate (2022)# | Year most recent published survey was conducted^ | Summary of findings |
|---|---|---|---|---|
| PICTs that have eliminated lymphatic filariasis as a public health problem | | | | |
| Cook Islands | 2016 | 15,406 | 2013–14 [17] | 2,903 participants from ten islands were tested. Only one person (an adult) was antigen (Ag) positive for LF. No Ag-positive individuals were found on the other 11 islands. The national Ag prevalence was calculated at 0.23%. |
| Niue | 2016 | 1,543 | 2009 [18] | A whole-population survey (n = 1,378) found an overall LF Ag prevalence of 0.5% with no cases in six to seven-year-old children. |
| Vanuatu | 2016 | 307,941 | 2010–12 [19] | 4,480 school-aged children were surveyed in three sampling units; no Ag-positive cases were found. Subsequently, a transmission assessment survey was conducted in one sampling unit that found two Ag-positive children out of 933 tested. The national Ag prevalence was calculated at 0.2%. |
| Wallis and Futuna | 2016 | 11,304 | 2012 [20] | A survey of 939 school-age children found three were Ag-positive cases, giving an overall Ag prevalence of 0.3%. |
| Palau | 2017 | 17,976 | 2001 [21] | A baseline assessment survey conducted in 2001 found nine Ag-positive cases (all from the same village) out of 2,031 people examined. The Ag prevalence was calculated at 0.4%. |
| Tonga | 2017 | 99,283 | 2015 [22] | The survey of 2,806 children from across all five administrative divisions found one Ag-positive case, giving an overall Ag prevalence of 0.04%. |
| Marshall Islands | 2017 | 54,446 | 2003 [21] | A 2001 baseline assessment survey of 2,003 people on two islands found two positive cases on Mejit Island, giving an Ag prevalence of 0.1%. In 2002, a blood antigen survey on Mejit and Alluk islands found 130 Ag-positive cases among 294 people (44.2%) and 71 Ag-positive cases among 244 (29%), respectively. A similar survey conducted in 2003 found no positive cases among 217 people examined on Wotje Atoll and the 318 people examined on Ebon Atoll. |
| Kiribati | 2019 | 122,735 | 1999–2000 [21] | A baseline assessment A survey found an Ag prevalence of 1.7%. |
| PICTs that have not yet eliminated lymphatic filariasis as a public health problem | | | | |
| American Samoa | - | 57,085 | 2016 [23, 24] | 135 of 2,671 survey participants were Ag-positive giving an Ag prevalence of 5.1%, confirming ongoing LF transmission in previously known clusters and hotspots, and identifying new potential hotspots that warrant investigation. |
| Fiji | - | 901,603 | 2007 [25] | A nationwide stratified cluster survey found the Ag prevalence to be 9.5%, ranging from 0.9% in the Western Division to 15.4% in the Eastern Division. Microfilaria (Mf) prevalence was 1.4%. |
| French Polynesia | - | 280,855 | 2008 [26] | A cross-sectional, stratified, three-cluster sampling study of inhabitants aged 2 years and older found an ICT-positive prevalence of 11.3% and Mf-prevalence of 10%. |
| Federated States of Micronesia | - | 105,987 | 2003 [27] | A survey of 233 participants on Satawal Islands found 96 (38%) were Ag-positive. 55 (22%) were found to have circulating Mf. |
| New Caledonia | - | 274,330 | 2013 [28] | A survey of 1,035 participants identified seven Ag-positive cases. All participants tested negative for Mf blood smears and filarial DNA. The overall LF Ag prevalence was 0.62% (95% CI [0.60–0.63]). Although this is below the WHO threshold for elimination, the absence of epidemiological evidence excluding potential domestic transmission has prompted health authorities to consider the possibility of ongoing disease circulation. |
| Papua New Guinea | - | 9,311,874 | Multiple years [29, 30] | Multiple surveys using different methods have been conducted in different locations. Graves et al. (2013) summary showed LF prevalences of 30.4–64.7% for the period 1983–92, 30.1–56.9% for the period 1993–2000, and 7.8–12.8% for the period 2003–11. |

*(Continued)*

**Table 1.** (Continued)

| Pacific island countries and territories (PICTs) | Year validated by WHO as having eliminated LF | Mid-year population estimate (2022)[#] | Year most recent published survey was conducted[^] | Summary of findings |
|---|---|---|---|---|
| Samoa | - | 200,999 | 2023 [31] | 623 participants aged >5 years from 125 randomly selected households in eight sampling units were surveyed. Ag-positive cases were found in six of the eight sampling units. The adjusted Ag prevalence was 9.9% (95% CI 3.5–21.0). |
| Tuvalu | - | 10,778 | 2004 [21] | A whole-population serosurvey found 973 Ag-positive cases among 8,173 people tested, giving an Ag-positive prevalence of 11.9%. |
| **PICTs where lymphatic filariasis is not endemic** | | | | |
| Guam (population: 179,900); Nauru (11,974); Northern Mariana Islands (56,986); Pitcairn Island (50), Solomon Islands (744,407); Tokelau (1,497). | | | | |

[#] Source: [32]. Population figures are mid-year estimates based on interpolating nations' census data and projections. The futures may differ from the actual census count in census years due to adjustments made by the Pacific Data Hub to allow comparability.

[^]The information presented is derived from publicly available sources. More recent but unpublished results may also exist. Additionally, some data pertain to specific sampling units or subsets within a country, and, as such, readers are advised to refer to the source document (cited) for a detailed account of the sampling frame used and results.

## Aim

This manuscript presents the study protocol for research that will use NGT to explore, synthesise, and prioritise the views of NTD program staff from LF-eliminated PICTs regarding the challenges faced in implementing surveillance for LF. Building on the main challenges identified, the study aims to develop a list of priority operational research questions that, once answered, will inform context-relevant policy and programmatic design. Further, the findings from this research are expected to inform national and regional LF guideline development, including the 'in draft' WHO guidelines for PVS of LF.

## Methods

### Participant recruitment and target sample size

A letter inviting participation in the NGT will be sent by email to national NTD program staff from LF-eliminated PICTs who are registered to attend the Coalition for Operational Research into Neglected Tropical Diseases (COR-NTD) meeting for the Pacific Islands in Brisbane, Australia on 25–26 September 2024. The letter will explain the study's objectives, rationale, risks, and benefits for participants. It will also provide a study participant information sheet and consent form and convey that the study has received ethical approval from The University of Queensland Human Research Ethics Committee. The timing of the NGT is opportunistic, as representatives from across the PICTs will be attending the COR-NTD meeting.

We aim to have one or two representatives from each PICT that has eliminated LF as a public health problem in the study (i.e., a purposefully selected sample of eight to 16). We propose to limit NGT discussion groups to a maximum of ten participants (i.e., if >10 participants elect to participate, two groups of NGT discussion will be formed).

### Eligibility criteria

To be eligible for the NGT, a participant must have at least two years' experience with their national NTD program and, during this time, have been involved in LF-related planning, surveillance, or response activities.

## Consent process

A study participant information sheet and consent form will be attached to the email sent to participants. The email will request that the consent form be returned before the NGT. Verbal confirmation of consent will be obtained at the start of NGT, and participants will be advised that they can withdraw their consent at any time without prejudice. Consent will also be obtained to audio-recorded the NGT session.

We do not anticipate that participants will experience harm or psychological distress because of participating in the NGT. Participants are health professionals who routinely engage in stakeholder consultation activities.

## Ethics

This study has the approval of The University of Queensland's Human Research Ethics Committee (Project# 2024/HE000224).

## Nominal group technique implementation

**Room set-up, equipment, and facilitation.**   The NGT will take place during a 2-hour face-to-face meeting. Participants will be arranged in a circle to ensure an egalitarian atmosphere, eliminating any sense of dominance or hierarchy. Each participant will receive a clipboard, adhesive note paper (i.e., a 'Post-it' notepad), and a pen. Additionally, a space on the wall will be designated for displaying and making participant-generated ideas visible to everyone.

One researcher will facilitate the NGT, guiding the process and providing explanations. A co-investigator will assist the facilitator as needed.

**Introducing the NGT process.**   At the start of the NGT, the facilitator will welcome the participants and invite them to introduce themselves, serving as an icebreaker. The facilitator will then spend 5–10 minutes explaining the purpose of the research and providing a step-by-step overview of the NGT process. This explanation will be supported by a visual aid and a simple example to ensure clarity. Participants will be invited to ask questions and seek clarification throughout this introductory phase.

**Step 1. Presentation of a nominal question, group members idea generation, clarification.**   To begin, the facilitator will present a nominal question verbally and on a poster for visibility. The question will be: "What do you consider to be the most significant challenges you face, or anticipate facing, in implementing PVS for LF in your setting?" Participants will be reminded that the aim is to gather the full range of responses, and that no response is wrong.

Participants will then have 2 minutes to reflect on the question and make individual notes. Following this, the facilitator will invite one participant to share one of their responses. This response will be clarified, recorded briefly on a Post-it note, and displayed on the wall. Participants will be limited to one response at a time to ensure dominant voices do not overpower others' ability to make contributions. The facilitator will continue to invite participants, one-by-one, to contribute one response at a time until all responses are captured. This may take several 'rounds' of idea generation.

Once all responses are displayed, participants will gather around the wall. The facilitator will prompt the group to identify overlapping responses and suggest possible merges. Participants will also be encouraged to seek clarification on any responses and revise wording if necessary. Participants will be invited to add any new ideas they may have, which will be incorporated into the display.

**Step 2. Individual ranking.**   The next step of the NGT involves participants reflecting on and ranking the responses generated by the group. This will be done individually using a

simple voting method developed by Delbecq [33], where each participant is allocated a set number of votes (e.g., ten) to distribute among the ideas as they choose. Participants may allocate all votes to one idea or distribute them across multiple ideas according to their perspectives.

The votes will be collected and tallied once all participants have completed the ranking. During this process, participants will be invited to take a 5-minute break. Upon their return, the facilitator will arrange the Post-it notes displaying the ideas in rank order. Ideas resonating most with the group are expected to appear higher in the order, while those considered less critical will rank lower. The facilitator will emphasise that the rankings are relative and that a lower rank does not imply the idea is unimportant.

The facilitator will then open a discussion about the rankings, asking questions such as, "Is this what you expected?" or "Would those who ranked this idea highest like to explain their reasoning?" If significant differences in opinion arise, the discussion will continue, and participants will have the option to repeat the ranking exercise if they are willing (i.e., conduct Step 2 again).

**Step 3. Scoping priory research questions related to highly ranked challenges.** The final step of the NGT will focus on generating operational research questions identified by national officers. The facilitator will invite participants to reframe the three highest-ranking responses to the nominal question into operational research questions. Given the uncertainty surrounding the nature of the emerging questions and the limited time available, a specific framework for drafting operational research questions is not proposed at this stage. Such refinement will occur at a later point.

When addressed, these questions will provide critical, context-specific insights to inform LF surveillance practices and strategy development across the PICTs. We recognise that identifying and refining operational research priorities, as well as securing political and financial commitments for action, is a time-intensive process. Therefore, we propose this stage of the NGT be the initial step towards developing a Pacific officer-led operational research agenda for LF.

**Analysis.** The ranking data will be quantitatively described, while discussions regarding the prioritisation, rationale for decision-making, and sentiments expressed will be qualitatively analysed using an inductive thematic approach, as outlined by Braun and Clarke [34], and Terry and colleagues [35]. Comparisons will be made between groups based on LF status, recency of LF surveillance activities, and demographic and economic factors, among other relevant variables.

## Discussion

This research aims to gather insights into the primary challenges faced by health officers in PICTs that have eliminated LF, particularly regarding PVS. Additionally, it will initiate discussions on the research priorities of national health officers related to LF surveillance, with the goal of generating knowledge that is directly relevant to PICTs' policy and practice for LF surveillance.

Consensus group methods are widely employed to identify and assess areas where evidence is incomplete [5, 36, 37]. While numerous consensus methods have been developed, including the Delphi technique, the Consensus Development Conference approach, and the Research and the Development/University of California Los Angeles (RAND/UCLA) Appropriateness method [38], NGT was chosen for this study as it facilitates rapid, time-bound group decisions. Its structure enables real-time feedback and controlled interaction, ensuring all voices in the group contribute equally. The participants in the NGT are experts in LF surveillance in their

relevant context, which positively influences the findings' validity, credibility, reliability, and acceptability [37]. Key advantages of NGT include the generation of numerous ideas in a relatively short timeframe [38], equal participation and prioritisation of ideas, 'ownership' of the outcomes, and immediate feedback [34]. Noted limitations include a relatively small number of participants and a facilitator, as well as the need to overcome logistical challenges, such as the need for participants to be in the same location [38].

To the best of our knowledge, this study represents the first application of a methodologically rigorous consensus group method to gather insights and identify operational research priorities for surveillance of LF, or any other disease, in the PICTs. Consequently, it will offer a novel approach to stakeholder consultation for end-user priority setting that will inform future program design. Specifically, the findings from this research are expected to inform national and regional LF PVS guideline development, including the 'in draft' WHO guidelines for PVS of LF.

## Author Contributions

**Conceptualization:** Adam T. Craig, Harriet Lawford, Colleen L. Lau.

**Methodology:** Adam T. Craig, Harriet Lawford, Satupaitea Viali, George Tuitama, Colleen L. Lau.

**Project administration:** Adam T. Craig, Harriet Lawford.

**Supervision:** Adam T. Craig, Colleen L. Lau.

**Writing – original draft:** Adam T. Craig, Harriet Lawford, Colleen L. Lau.

**Writing – review & editing:** Adam T. Craig, Harriet Lawford, Satupaitea Viali, George Tuitama, Colleen L. Lau.

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
