## [Decision Letter · Decision Letter 0]

21 Oct 2024

PONE-D-24-41211Challenges in implementation of post-validation surveillance for lymphatic filariasis in the Pacific Islands: A nominal group technique-based research study protocolPLOS ONE

Dear Dr. Craig,

Thank you for submitting your manuscript to PLOS ONE. After careful consideration, we feel that it has merit but does not fully meet PLOS ONE’s publication criteria as it currently stands. Therefore, we invite you to submit a revised version of the manuscript that addresses the points raised during the review process.

**ACADEMIC EDITOR:**

**I have invited five experts to review the manuscript. Of them, three agreed to review the ms. All the three reviewers considered the protocol is timely and relevant in the context of post-validation surveillance for lymphatic filariasis. But also raised both major and minor comments. I would request the authors to focus on the major comments on the methodology related to how to elicit meaningful responses from participants or how program personnel would able to identify challenges and provide insights especially in the absence of clear guidance from WHO or other stake holders.**

We look forward to receiving your revised manuscript.

Kind regards,

Swaminathan Subramanian, Ph.D.

Academic Editor

PLOS ONE

**Journal Requirements:**

2. In the ethics statement in the Methods, you have specified that verbal consent will be obtained. Please provide additional details regarding how this consent will be documented and witnessed, and state whether this was approved by the IRB.

4. We note you have included a table to which you do not refer in the text of your manuscript. Please ensure that you refer to Table 1 in your text; if accepted, production will need this reference to link the reader to the Table.

Reviewers' comments:

Reviewer's Responses to Questions

**Comments to the Author**

1. Does the manuscript provide a valid rationale for the proposed study, with clearly identified and justified research questions?

Reviewer #1: Partly

Reviewer #2: Yes

Reviewer #3: Yes

2. Is the protocol technically sound and planned in a manner that will lead to a meaningful outcome and allow testing the stated hypotheses?

Reviewer #1: Partly

Reviewer #2: Yes

Reviewer #3: Yes

3. Is the methodology feasible and described in sufficient detail to allow the work to be replicable?

Reviewer #1: Yes

Reviewer #2: Yes

Reviewer #3: Yes

4. Have the authors described where all data underlying the findings will be made available when the study is complete?

Reviewer #1: Yes

Reviewer #2: Yes

Reviewer #3: Yes

5. Is the manuscript presented in an intelligible fashion and written in standard English?

Reviewer #1: Yes

Reviewer #2: Yes

Reviewer #3: Yes

6. Review Comments to the Author

You may also provide optional suggestions and comments to authors that they might find helpful in planning their study.

**Reviewer #1:** Comments on the paper titled Challenges in implementation of post-validation surveillance for lymphatic filariasis in

the Pacific Islands: A nominal group technique-based research study protocol

Post-validation surveillance (PVS) is an important component of the national Lymphatic filariasis (LF) elimination programs. The progress of PVS is tardy, so far, in the countries that have eliminated LF as a public health problem. In this context, the objective of the authors to identify the challenges in implementing PVS and operational research areas in Pacific island countries is timely and relevant.

As the authors noted (Line 13), there is limited guidance from WHO or other stakeholders regarding the components and strategies of post-validation surveillance (PVS). Many programs, whether in the PVS phase or post-MDA surveillance phase, may lack a clear understanding of how to organize and implement PVS or estimate its costs. In this context, it is unclear how the authors expect to elicit meaningful responses from participants or how program personnel would be able to identify challenges and provide insights, especially in the absence of concrete guidance. For instance, if I were a program manager or deputy with little knowledge of PVS strategies, it would be difficult to identify challenges or offer perspectives. The authors should address this gap in detail.

The other points are as follows:

Line 11: It is not 'eradicated'; it is 'eliminated'

Lines 56-60: The figures 120 million people and 40 million people refer to the years prior to launching the LF elimination program. The current figures are much lower due to the impact of the LF elimination programs. The authors should provide either current figures or correct the sentence to indicate the figures pertain to pre-GPELF period.

Line 68: threshold of 1% - provide whether it is Ag prevalence or Mf prevalence.

Line 93: COR NTD meeting: The dates and venue details of the meeting should be provided.

**Reviewer #2:** The manuscript is technically sound, and the authors have well described the protocol for a NGT-based research study addressing the pertinent issue of challenges in the implementation of post validation surveillance for LF.

There are a few minor clarifications and suggestions that may be addressed by the authors:

Line no. 52: "potential for dominant voices to influence group decisions"- how this will be addressed in the study shall be briefly explained in the Methods section.

Line nos. 56-60: Reference no. 8 is quite old. Recent figures and reference shall be provided.

Line no. 67: "..prevalence of LF to below the prevalence threshold of 1%.."- I suppose it means Mf prevalence<1%. The sentence shall be reframed for better clarity.

Line nos. 61-71: The paragraph discusses about the LF situation in the PICTs. A line on the parasite (W.b/ B.m) and vector species involved in transmission in PICTs can be included for better understanding of the readers.

Line nos. 95-98 and Line nos. 108-110, 117-118 are repeated.

Line no. 128: A sentence on consent for audio recording shall be included under "Consent process"

Line no. 189: Under Discussion, the authors can briefly discuss the advantages of NGT over Delphi method and any other consensus method available.

Line no. 198 is unclear and shall be reframed: co-design what?

Line no. 200: :"..in draft WHO guidelines for PVS of LF"- the same shall be mentioned in Introduction after line no: 79-80

**Reviewer #3:** Comments on the paper entitled “Challenges in implementation of post-validation surveillance for lymphatic filariasis in the Pacific Islands: A nominal group technique-based research study protocol”

PLOS ONE accepts submissions for publication of Study Protocols for any study type within the journal’s scope. The proposed study Protocols describe detailed plans for conducting research, including the background, rationale, objectives, methodology, statistical plan, and organization of a research project. This protocol can be considered for publication as the study plans to use consensus group method by following nominal group technique and bring out the challenges faced in implementing surveillance for LF and research priorities for LF LF-PVS, with an ultimate objective of developing national and regional guidelines.

Abstract

1. Line 11: replace “eradicated the disease” by “eliminated”

2. Line 12: replace “prevent the” by “verify absence of resurgence and …..”

3. Line 30: add “post validation” before “surveillance”

4. Line 31: add Nominal Group technique

Introduction

5. Line 39: remove “method”

6. Line 64: add “in 16 countries” after (WHO)

7. Line 67-68: add after have “cleared Transmission Assessment Survey” and delete “reduced the prevalence of LF to below the prevalence threshold of 1%”.

8. Line 70: before “to” add “achieve elimination of LF” and delete eliminate

9. Line 73: replace figure1 by table 1

10. Table 1: Add a column “population” before “Year validated by WHO as having eliminated LF”, and give the latest population

Methods

11. Line 89: is it not biased to include NTD programme staff from LF-eliminated PICTs who are registered to attend the Coalition for Operational Research into Neglected Tropical Diseases (COR-NTD) meeting for the Pacific Islands, scheduled in September 2024?.

12. Line 102: rephrase as “two groups of NGT discussion will be formed”

13. Line 107: consent procedure is a repeat of lines 95-98. The text in lines 95-98 can be deleted.

14. Line 123-125: Additionally, a space on the wall will be designated for displaying and making participant-generated ideas visible to everyone. Should it mean flip chart?

15. Line 140: does it refer post validation surveillance?

Discussion

16. Line 192: does it mean post validation surveillance?

7. PLOS authors have the option to publish the peer review history of their article (what does this mean?). If published, this will include your full peer review and any attached files.

Reviewer #1: No

Reviewer #2: No

Reviewer #3: **Yes: **Krishnamoorthy Kaliannagounder

---

## [Author Response · Author response to Decision Letter 0]

22 Oct 2024

Reviewer #1:

Post-validation surveillance (PVS) is an important component of the national Lymphatic filariasis (LF) elimination programs. The progress of PVS is tardy, so far, in the countries that have eliminated LF as a public health problem. In this context, the objective of the authors to identify the challenges in implementing PVS and operational research areas in Pacific island countries is timely and relevant.

-- Noted

As the authors noted (Line 13), there is limited guidance from WHO or other stakeholders regarding the components and strategies of post-validation surveillance (PVS). Many programs, whether in the PVS phase or post-MDA surveillance phase, may lack a clear understanding of how to organize and implement PVS or estimate its costs. In this context, it is unclear how the authors expect to elicit meaningful responses from participants or how program personnel would be able to identify challenges and provide insights, especially in the absence of concrete guidance. For instance, if I were a program manager or deputy with little knowledge of PVS strategies, it would be difficult to identify challenges or offer perspectives. The authors should address this gap in detail.

-- While the absence of global guidance on PVS implementation may have impeded understanding and commitment, it has not been our experience that NTD program managers and staff are unaware of the rationale or epidemiological principles underlying PVS. Many Pacific Ministry of Health staff have extensive experience with LF elimination, having worked tirelessly to reduce LF prevalence significantly in their countries. Given their role, experience with LF and other NTD program delivery, and insights into the leavers that drive national policy and program decision-making processes, we believe these individuals are the most appropriate and among the best equipped to offer meaningful insights into the challenges faced in the delivery of PVS the Pacific context.

The other points are as follows:

Line 11: It is not 'eradicated'; it is 'eliminated'

-- Our mistake, thank you. The word has been changed. 

Lines 56-60: The figures 120 million people and 40 million people refer to the years prior to launching the LF elimination program. The current figures are much lower due to the impact of the LF elimination programs. The authors should provide either current figures or correct the sentence to indicate the figures pertain to pre-GPELF period.

-- The text has been updated and now reads, “Lymphatic filariasis has affected about 882 million people, primarily in tropical and subtropical regions of Africa, Southeast Asia, the Pacific Islands, and the Americas [10]. As of 2018, 51 million people were infected – a 74% decline since the start of WHO’s Global Programme to Eliminate Lymphatic Filariasis in 2000 [11]. Since 2000, over 9.7 billion treatments for LF have been administered through mass drug administration (MDA) campaigns, reaching more than 943 million people. In 2023, the number of people that require MDA was 657 million [12].” (around line 58)

1.5. Line 68: threshold of 1% - provide whether it is Ag prevalence or Mf prevalence.

-- The text has been updated to read, “reduced the prevalence of LF to below the microfilaria-positive prevalence threshold of 1%.” (around line 65).

1.6. Line 93: COR NTD meeting: The dates and venue details of the meeting should be provided.

-- The text has been updated to read “in Brisbane, Australia, on 25-26 September 2024.” (around line 89)

Reviewer #2: 

The manuscript is technically sound, and the authors have well described the protocol for a NGT-based research study addressing the pertinent issue of challenges in the implementation of post validation surveillance for LF.There are a few minor clarifications and suggestions that may be addressed by the authors:

2.1. Line no. 52: "potential for dominant voices to influence group decisions"- how this will be addressed in the study shall be briefly explained in the Methods section.

-- At around line 142, we have edited the text to read, “Participants will be limited to one response at a time to ensure dominant voices do not overpower others’ ability to make contribution. The facilitator will continue to invite participant, one-by-one,to contribute one response at a time until all responses are captured. This may take several ‘rounds’ of idea generation.”

2.2. Line nos. 56-60: Reference no. 8 is quite old. Recent figures and reference shall be provided.

This passage and reference have been replaced with the following. ““Lymphatic filariasis has affected about 882 million people, primarily in tropical and subtropical regions of Africa, Southeast Asia, the Pacific Islands, and the Americas [10]. As of 2018, 51 million people were infected – a 74% decline since the start of WHO’s Global Programme to Eliminate Lymphatic Filariasis in 2000 [11]. Since 2000, over 9.7 billion treatments for LF have been administered through mass drug administration (MDA) campaigns, reaching more than 943 million people. In 2023, the number of people that require MDA was 657 million [12].” (around line 58)

Line no. 67: "..prevalence of LF to below the prevalence threshold of 1%.."- I suppose it means Mf prevalence<1%. The sentence shall be reframed for better clarity.

-- The text has been revised and now reads, “reduced the prevalence of LF to below the microfilaria-positive prevalence threshold of 1%.” (around line 65).

Line nos. 61-71: The paragraph discusses about the LF situation in the PICTs. A line on the parasite (W.b/ B.m) and vector species involved in transmission in PICTs can be included for better understanding of the readers.

-- In response, the following text has been added around line 52. “Lymphatic filariasis (LF) is a neglected tropical disease (NTD) caused by three species of filarial worm: Wuchereria bancrofti, Brugia malayi, and B. timori and transmitted between humans by a range of mosquito species including Aedes, Culex, and Anopheles [7, 8]. W. bancrofti is the only filarial worm species found in the Pacific Islands.”

Line nos. 95-98 and Line nos. 108-110, 117-118 are repeated.

-- The second and third mentions of the letter of invitation have been removed, and the first instance (around line 90) edited to read, “A letter inviting participation in the NGT will be sent by email to national NTD program staff from LF-eliminated PICTs who are registered to attend the Coalition for Operational Research into Neglected Tropical Diseases (COR-NTD) meeting for the Pacific Islands in Brisbane, Australia on 25-26 September 2024. The letter will explain the study's objectives, rationale, risks, and benefits for participants. It will also provide a study participant information sheet and consent form, and convey that the study has received ethical approval from The University of Queensland Human Research Ethics Committee. The timing of the NGT is opportunistic, as representatives from across the PICTs will be attending the COR-NTD meeting.”

Line no. 128: A sentence on consent for audio recording shall be included under "Consent process"

-- The statement has been moved to the consent section (around line 118) and text tweaked for clarity. It now reads, “Consent will also be obtained to audio-recorded the NGT session.”

Line no. 189: Under Discussion, the authors can briefly discuss the advantages of NGT over Delphi method and any other consensus method available.

-- In response, from around line 213, we have added the following. “Consensus group methods are widely employed to identify and assess areas where evidence is incomplete [5, 35, 36]. While numerous consensus methods have been developed, including the Delphi technique, the Consensus Development Conference approach, and the Development/University of California Los Angeles (RAND/UCLA) Appropriateness method [36], NGT was chosen for this study as it facilitates rapid, time-bound group decisions. Its structure enables real-time feedback and controlled interaction, ensuring all voices in the group contribute equally. The participants in the NGT are experts in LF surveillance in their relevant context, which positively influences the findings' validity, credibility, reliability, and acceptability [36]. Key advantages of NGT include the generation of numerous ideas in a relatively short timeframe [37], equal participation and prioritisation of ideas, ‘ownership’ of the outcomes, and immediate feedback [35]. Noted limitations include a relatively small number of participants and a facilitator, as well as the need to overcome logistical challenges, such as the need for participants to be in the same location [37].”

Line no. 198 is unclear and shall be reframed: co-design what?

-- Thank you for the comment. The wording has been changed to “… inform future program design.”

Line no. 200: :"..in draft WHO guidelines for PVS of LF"- the same shall be mentioned in Introduction after line no: 79-80

The sentence, “Further, the findings from this research are expected to inform national and regional LF guideline development, including the 'in draft' WHO guidelines for PVS of LF” has been added to the aims section, around line 87 on the marked up version of the revised manuscript.

Reviewer #3: 

Comments on the paper entitled “Challenges in implementation of post-validation surveillance for lymphatic filariasis in the Pacific Islands: A nominal group technique-based research study protocol”

PLOS ONE accepts submissions for publication of Study Protocols for any study type within the journal’s scope. The proposed study Protocols describe detailed plans for conducting research, including the background, rationale, objectives, methodology, statistical plan, and organization of a research project. This protocol can be considered for publication as the study plans to use consensus group method by following nominal group technique and bring out the challenges faced in implementing surveillance for LF and research priorities for LF LF-PVS, with an ultimate objective of developing national and regional guidelines.

Abstract

1. Line 11: replace “eradicated the disease” by “eliminated”

-- this change has been made. Apologies for this obvious error. (around line 11)

2. Line 12: replace “prevent the” by “verify absence of resurgence and …..”

-- The suggested change has been made. (around line 12)

3. Line 30: add “post validation” before “surveillance”

-- the keyword “post-validation surveillance” has been added to the list of terms.

4. Line 31: add Nominal Group technique

-- The keyword “Nominal Group Technique” has been added to the list of terms.

Introduction

5. Line 39: remove “method”

-- the word “method” has been removed. (around line 39)

6. Line 64: add “in 16 countries” after (WHO)

-- the sentence has been updated to read, “The GPELF’s main objective was to achieve the elimination of LF as a public health problem across 73 endemic nations, , including 16 in the Pacific Islands region, by 2020 [13].” (around line 64)

7. Line 67-68: add after have “cleared Transmission Assessment Survey” and delete “reduced the prevalence of LF to below the prevalence threshold of 1%”.

8. Line 70: before “to” add “achieve elimination of LF” and delete eliminate

-- The suggested change has been made. (around line 72).

9. Line 73: replace figure1 by table 1

-- The figure heading of has been changed ‘table.’

10. Table 1: Add a column “population” before “Year validated by WHO as having eliminated LF”, and give the latest population

-- The suggestion has been accepted and the table modified accordingly. We report population data as done by the Pacific Data Hub and include the following footnote to the table. “Source: [28]. Population figures are mid-year estimates based on interpolating nations’ census data and projections. The futures may differ from the actual census count in census years due to adjustments made by the Pacific Data Hub to allow comparability.”

Methods

11. Line 89: is it not biased to include NTD programme staff from LF-eliminated PICTs who are registered to attend the Coalition for Operational Research into Neglected Tropical Diseases (COR-NTD) meeting for the Pacific Islands, scheduled in September 2024?.

-- As the proposed research is not experimental, we seek input from purposefully selected officers with experience and insights to share.

12. Line 102: rephrase as “two groups of NGT discussion will be formed”

-- The suggested wording has been adopted. (around line 115)

13. Line 107: consent procedure is a repeat of lines 95-98. The text in lines 95-98 can be deleted.

-- This has been addressed, as per a response to a comment above. See the revised consent section, around line 120.

14. Line 123-125: Additionally, a space on the wall will be designated for displaying and making participant-generated ideas visible to everyone. Should it mean flip chart?

-- No, we mean wall space as stated.

15. Line 140: does it refer post validation surveillance?

-- Yes. The sentence has been revised and now reads, “The question will be: “What do you consider to be the most significant challenges you face, or anticipate facing, in implementing PVS for LF in your setting?” (around line 154)

Discussion

16. Line 192: does it mean post validation surveillance?

-- Yes. The sentence has been revised and now reads, “Specifically, the findings from this research are expected to inform national and regional LF PVS guideline development…” (around line 216)

---

## [Decision Letter · Decision Letter 1]

4 Nov 2024

Perspectives on the implementation of post-validation surveillance for lymphatic filariasis in the Pacific Islands: A Nominal Group Technique-based study protocol

PONE-D-24-41211R1

Dear Dr. Craig,

We’re pleased to inform you that your manuscript has been judged scientifically suitable for publication and will be formally accepted for publication once it meets all outstanding technical requirements.

Kind regards,

Swaminathan Subramanian, Ph.D.

Academic Editor

PLOS ONE

Additional Editor Comments (optional):

Dear Authors,

We are pleased to inform you that all the three reviewers have recommended to accept the ms for publication in PLOS ONE (review reports attached). However, there are minor corrections from one of the reviewers (Reviewer 3). I would request you to take care of the minor corrections while submitting the final version.

Best wishes

Reviewers' comments:

Reviewer's Responses to Questions

**Comments to the Author**

1. Does the manuscript provide a valid rationale for the proposed study, with clearly identified and justified research questions?

Reviewer #1: Partly

Reviewer #2: Yes

Reviewer #3: Yes

2. Is the protocol technically sound and planned in a manner that will lead to a meaningful outcome and allow testing the stated hypotheses?

Reviewer #1: Partly

Reviewer #2: Yes

Reviewer #3: Yes

3. Is the methodology feasible and described in sufficient detail to allow the work to be replicable?

Reviewer #1: Yes

Reviewer #2: Yes

Reviewer #3: Yes

4. Have the authors described where all data underlying the findings will be made available when the study is complete?

Reviewer #1: Yes

Reviewer #2: Yes

Reviewer #3: Yes

5. Is the manuscript presented in an intelligible fashion and written in standard English?

Reviewer #1: Yes

Reviewer #2: Yes

Reviewer #3: Yes

6. Review Comments to the Author

You may also provide optional suggestions and comments to authors that they might find helpful in planning their study.

Reviewer #1: Comments on Perspectives on the implementation of post-validation surveillance for lymphatic filariasis in the Pacific Islands: A Nominal Group Technique-based study protocol

The contribution of PIC to the field of epidemiology and the control of lymphatic filariasis (LF) since the 1960s, as well as to LF elimination efforts since 2000, is indeed significant and commendable. However, this history alone does not necessarily establish PIC or any other region as “among the best equipped to offer meaningful insights into the challenges faced in the delivery of PVS in the Pacific context.” Without a well-defined strategy, discussing effective delivery and associated challenges seems premature. Therefore, the authors’ argument lacks sufficient support. That said, I appreciate the enthusiasm and dedication of the authors, and I encourage them to publish and implement their protocol.

Reviewer #2: The authors have addressed all comments in the revised version of manuscript. The necessary changes made are appropriate and acceptable.

Reviewer #3: This paper may be accepted for publication in the PLOS One Journal as the author have responded to the comments of reviewer 3. However, the authors need to respond to the following two comments:

Previous comment 6:. Line 64: add “in 16 countries” after (WHO):

Authors’ response: the sentence has been updated to read, “The GPELF’s main objective was to achieve the elimination of LF as a public health problem across 73 endemic nations, including 16 in the Pacific Islands region, by 2020 [13].” (around line 64)

Present comment: Line 69-71 of the revised manuscript: Correct the sentence “The GPELF’s main objective was to achieve the elimination of LF as a public health problem across 73 endemic nations, including 16 in the Pacific Islands region, by 2020 [15] [16]]” as “The GPELF’s main objective was to achieve the elimination of LF as a public health problem across 72 endemic nations, including 16 in the Pacific Islands region, by 2020 [15]. [16]

Previous comment 7: Line 67-68: add after have “cleared Transmission Assessment Survey” and delete

“reduced the prevalence of LF to below the prevalence threshold of 1%”.

The authors are yet to respond to this comment.

7. PLOS authors have the option to publish the peer review history of their article (what does this mean?). If published, this will include your full peer review and any attached files.

Reviewer #1: No

Reviewer #2: No

Reviewer #3: **Yes: **Krishnamoorthy Kaliannagounder

---

## [Editor Report · Acceptance letter]

18 Nov 2024

PONE-D-24-41211R1 

PLOS ONE

Dear Dr. Craig, 

I'm pleased to inform you that your manuscript has been deemed suitable for publication in PLOS ONE. Congratulations! Your manuscript is now being handed over to our production team.

Kind regards, 

on behalf of

Dr. Swaminathan Subramanian 

Academic Editor

PLOS ONE